# Assessment of the Impact of Land Use Change on Spatial Differentiation of Landscape and Ecosystem Service Values in the Case of Study the Pearl River Delta in China

**Ren Yang \*, Baoqing Qin and Yuancheng Lin**

School of Geography and Planning, Sun Yat-sen University, Guangzhou 510275, China; qinbq@mail2.sysu.edu.cn (B.Q.); linych37@mail2.sysu.edu.cn (Y.L.)
**\*** Correspondence: yangren666@mail.sysu.edu.cn

**Abstract:** Industrialization and urbanization have led to continuous urban development. The rapid change in land-use type and extent has a significant impact on the capacity of ecosystem services. Changes in the landscape pattern of roads, rivers, railway stations, and expressway entrances and exits have evident geographical proximity effects. We used landscape pattern indices and ecosystem service value (ESV) to evaluate the landscape pattern and ESV spatial differentiation of the Pearl River Delta region and its typical transportation infrastructure and rivers in 1990, 2000, and 2017. The results show that rapid urbanization and industrialization have led to changes in urban land use along the Pearl River Estuary. Urban land changes on the east bank of the Pearl River are greater than urban land changes on the west bank of the Pearl River; the landscape diversity of the Pearl River Delta has increased, the connectivity of the landscape has decreased, and the degree of fragmentation has increased. Second, the city size of the Pearl River Delta was negatively correlated with the ESVs. The ESVs in the core areas of the Pearl River Delta urban agglomeration were smaller than those in the fringe areas. With the gradient change in urban land use, ESVs showed a growing trend from the city center to the surrounding areas. The key areas for ecological protection and restoration should be central urban areas and suburbs. Third, the siphoning effect of the buffer zones of railway stations and expressway entrances and exits was very strong and drove the development and utilization of the surrounding land. As the degree of land development in the buffer zone decreased, the ESVs increased. Fourth, different grades of roads in the Pearl River Delta had different impacts on the regional landscape and ESVs. County roads had a greater interference effect than expressways, national roads, and provincial roads, and the riverside plains of the Pearl River Delta have a large development space, low urban development costs, and multiple land-use activities that have profoundly changed the landscape of the river buffer zone.

**Keywords:** land-use change; landscape index; ecosystem service value; temporal variation; Pearl River Delta

## 1. Introduction

With the rapid advancement of urbanization, the urban population is growing rapidly around the world, especially in developing countries in Asia and Africa. In recent years, China's urban population has grown rapidly, and the urbanization rate has risen from 10.64% in 1949 to 63.89% in 2020. However, rapid urbanization has prompted drastic changes in landscape patterns around cities. In particular, impervious surfaces, such as buildings and roads continue to invade natural landscapes, causing the loss of ecosystem services, such as food production and biodiversity decline, as well as air, water, and soil pollution [1,2]. The change in land-use caused by urbanization has led to changes in the landscape pattern, which in turn affects the level of ecosystem services (ES) and attracts worldwide attention. Specifically, the transformation of the landscape pattern of land-use will cause changes in the material flow, energy flow, and information flow of

the surrounding ecosystems, leading to changes in ecosystem types, and positively or negatively impacting the urban ecological balance and sustainable development. Therefore, analyzing the changes in land-use landscape patterns can reveal spatial trends of urban development and human activities [3,4].

The landscape pattern is a heterogeneous entity comprising the spatial mosaics of interacting ecosystems. These changes can directly reflect the characteristics of regional social, economic, and ecological changes and reveal the relationship between human activities and the natural environment. Therefore, it is important to reveal the relationship between human activities and the natural environment [5]. In 1939, German eco-geographer Troll first proposed the concept of "landscape ecology", then Forman et al. proposed the "patches-corridor-mechanism" landscape composition model, which has a profound impact on subsequent landscape ecology [6]. Ecosystem services refer to the life-sustaining products and services obtained directly or indirectly through the structure, processes, and functions of the ecosystem, which form and maintain the environmental conditions and utility of human survival and development [7]. Ecosystem services include four service elements of supply, regulation, culture and auxiliary. Among them, the supply service refers to the provision of ecological products for nature, the regulation service is to regulate the ecological environment, the cultural service is the cultural and aesthetic enjoyment provided by the nature, and the auxiliary service includes the auxiliary function to the formation and development of land and plants [8–10]. Costanza et al. proposed using the ecosystem's economic value per unit area to assess ecosystem service value (ESV) [11]. The value evaluation method is simple to operate, and the results are intuitive and clear [12–14]. Xie et al. revised Costanza's evaluation method of global ecosystem service function value and formulated a value table per unit area for different terrestrial ecosystems adapted to China's national conditions [15].

Scholars have researched the impact of land-use landscape patterns and ESV changes at different areal types [16–18]. In metropolitan areas, Kremer et al. assessed and analyzed how the value of various ecosystem services of New York City's urban green infrastructure changed with the change in governance priorities [19]. Ye et al. quantified land use and ESV changes in the Guangzhou-Foshan metropolitan area and advocated decision-making analysis to support urban planning to achieve the sustainability of ecosystem services [20]. In intensive agricultural areas, the formulation of agricultural policies and the improvement of technology will affect the homogenization of the watershed landscape. The homogenization of landscape patterns in most areas co-occurred with the diversification of some landscape features in some areas [21]. In ecologically sensitive areas, the evaluation of the ESV change can provide guidance for the sustainable use and management of regional resources, thereby protecting significant, but fragile ecosystems [22]. Moreover, with the growth of population in surrounding areas and the continuous expansion of land use demand, the integrity and ecological health of the natural reserve are challenged [23]. In short, the research on the impact of land-use landscape patterns and ESV changes has mostly been based on remote sensing images, GIS technology, landscape pattern indices, and ecosystem service value evaluation methods. Moreover, it has mainly been carried out in towns and watersheds, focusing on rapid urbanization areas and fragile ecological environments.

In addition, the land-use landscape pattern change along with the transportation infrastructure and rivers and their ecological impact have also received much attention from society and academia. Transportation infrastructure is indispensable in urban and rural areas, and it plays an important role in expanding the scope of human activities, promoting social and economic development and urbanization. With the expansion of transportation infrastructure, the influence of human activities expands. For example, Asher et al. discussed the impact of country road facilities construction on the forest ecological environment, and the results showed that the facility construction has no direct impact on forest quality, but the upgrading of roads has caused changes in ecological corridors, which have led to adverse environmental impacts [24]. The continuous expansion

of human activities puts tremendous pressure on the ecological environment within the radiation range of transportation infrastructure. Rivers are a crucial part of the landscape and ecosystem in urban and rural areas, they provide regulation, supply, and cultural ecosystem services for areas within the basin, including flood risk regulation, crop production, and daily recreation [25]. The construction of expressways changed the land-use types along the road, leading to changes in the ecological service functions and the area, connectivity and material of the road have a direct impact on the surrounding waters and land ecology [26]. Much research has been conducted on the changes in landscape patterns and ESVs of land use around a single road, road network, or river. However, there has been little literature on land-use landscape pattern changes in integrated transportation networks. Moreover, while numerous research has examined the effects of land use on landscape pattern changes and the ecological impact of the transportation infrastructure within a single city [27], few have combined cross-regional transportation infrastructure and rivers to explore the spatiotemporal evolution of landscape patterns and ESV changes in transportation infrastructure and river buffer zones. Therefore, it would be beneficial to evaluate the land-use landscape change of regional transportation infrastructure and rivers quantitatively and study the spatial heterogeneity of the landscape pattern and ESVs of typical transportation infrastructure and river buffer zones. It would provide theoretical support for regional ecological security.

As one of the most developed regions in China, the Pearl River Delta (PRD) region has experienced rapid socio-economic development and urbanization, with the rapid expansion of regional construction and major destruction of natural landscapes [28]. Road traffic infrastructure and natural water systems form the skeleton on which the economic development of the PRD region has grown, affecting the regional landscape pattern and ecology. Evaluating the changes in ESV caused by changes in the land-use pattern in the area and typical transportation infrastructure will strengthen the understanding of the relationship between urban development and the regional ecological environment. Therefore, this study takes the PRD region as an example to quantify the landscape pattern and ESV changes in the region's overall and typical transportation infrastructure and river multi-ring buffer zones in 1990, 2000, and 2017. The main research goals are: (1) analyze the overall landscape pattern and ESV of the Pearl River Delta; (2) explore the typical transportation infrastructure and landscape pattern along the river and the ESV spatial differentiation characteristics based on the buffer zone and (3) analyze the evolution of typical transportation infrastructure and river landscape patterns.

## 2. Materials and Methods

### 2.1. Study Area

The PRD region (112°–115°5 E, 21°5–25° N) is located in the south-central part of Guangdong Province, China, including Guangzhou, Shenzhen, Dongguan, Foshan, Zhuhai, Huizhou, Zhongshan, Jiangmen, and Zhaoqing (Figure 1). It is adjacent to Hong Kong and Macau, with a total area of approximately 55,369 km$^2$. The region is formed by the accumulation of sediment brought by the Xijiang, Beijiang, Dongjiang rivers, and the tributaries Zengjiang, Tanjiang, and Suijiang in the Pearl River system. The climate of the Pearl River Delta is mild and humid, and it is a typical subtropical climate. The average annual temperature is between 21 °C and 23 °C, and the high-temperature season is synchronous with the rainy season. It has fertile land resources, mainly red soil, which is suitable for agricultural development. The vegetation in the Pearl River Delta is mainly subtropical evergreen broad-leaved vegetation. This area was at the forefront of China's reform and opening up and has been declared a national optimized development zone [29]. In 2017, the total GDP was $7.58 \times 10^{12}$ Chinese Yuan (CNY), and the urbanization level was 85.29%, much higher than the national urbanization level of 59.58%. In 2017, the scale of built-up land in the PRD reached 7666.96 km$^2$, accounting for 12.62% of the total land use. The cropland area was 12,125.17 km$^2$, accounting for 19.96%, and the waterbody area was 3724.62 km$^2$, accounting for 6.13%. The PRD has a high population density. The

permanent population was 61,505,400, accounting for 55.07% of the permanent population in Guangdong Province, in 2017. The PRD has a comprehensive transportation network, with a total highway mileage exceeding 63,630 km. Since 1990, the PRD has rapidly become urbanized. As a result, the land-use structure has undergone tremendous changes, and it has become a key area for land-use change research in China [30–33]. Exploring the spatial differentiation characteristics of land use in the landscape and the ESV of typical transportation infrastructure in the PRD region could have important practical significance for land consolidation and sustainable use of land resources in the urbanization process.

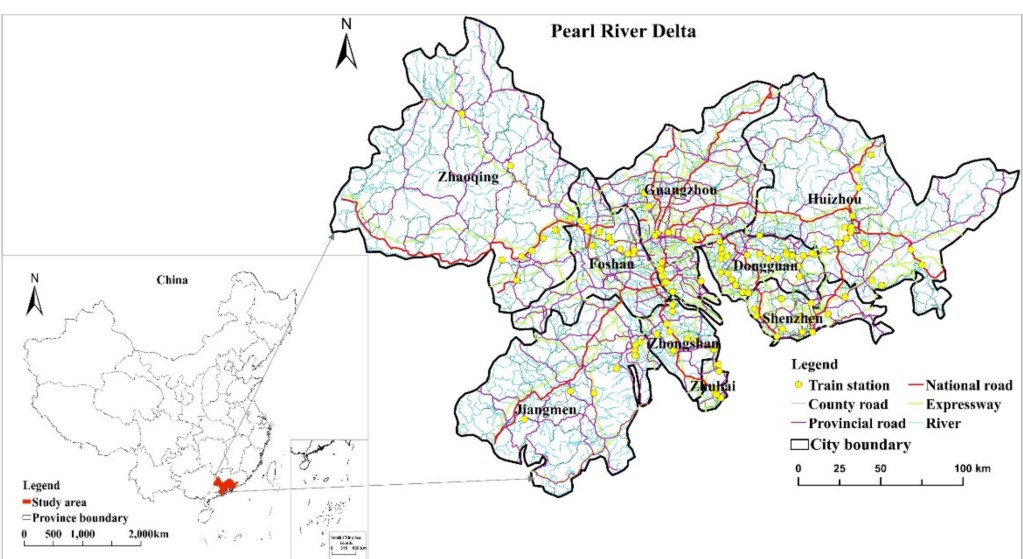

**Figure 1.** Location of the PRD region.

*2.2. Data Sources*

The land-use data from 1990, 2000, and 2017 used in this study were obtained from the Resource and Environmental Science Data Center of the Chinese Academy of Sciences (http://www.resdc.cn, accessed on 25 March 2021) with a resolution of 30 m × 30 m. According to the research results of land use classification [34], the data included cropland, woodland, grassland, waterbody, built-up land, unused land. Considering that the administrative divisions of the PRD have not changed much after 2000, and the typical traffic vector data for 2020 is not available, this study could not be updated to the latest year, and only three periods of comparison were made. Based on the "Statistical Yearbook of Guangdong Province" from 1991 to 2018, this study collected data on the total output and planting area of grain crops in the PRD and used it to calculate ecosystem service value. The road vector data were obtained from the Geospatial Data Cloud (http://www.gscloud.cn/, accessed on 25 March 2021). The vector data on railway stations, expressway entrances and exits, rivers, and cities were derived from the National Catalogue Service for Geographic Information (http://www.webmap.cn/main.do?method=index, accessed on 25 March 2021).

*2.3. Methods*

2.3.1. Buffer Analysis Method

The buffer is a polygonal layer based on points, lines, and surfaces [35]. Buffer analysis is an important spatial analysis tool used to solve the problem of geographical proximity. In this study, roads, rivers, stations, expressway entrances and exits, and cities were selected as buffer centers. With the support of ArcGIS10.3 software, a buffer zone was established every 1 km to generate a 10-level buffer zone. The buffer zones were then superimposed with the land-use maps of the PRD in 1990, 2000, and 2017 for subsequent data processing.

### 2.3.2. Landscape Index Analysis

In landscape ecology, the landscape index analysis method is used to analyze and understand the basic pattern characteristics and evolution laws of land-use change [36]. The current land-use data is mainly a discontinuous variable, and the landscape index can highly condense landscape pattern information, reflecting the basic characteristics of landscape structure composition and spatial configuration. Therefore, the landscape index is widely used to describe the evolution of landscape patterns and their impact on ecological processes quantitatively [37]. We studied existing research results and considering the actual situation in the PRD, decided to conduct our research at the landscape level [31,38]. In this study, landscape fragmentation index, shape index, aggregation index, and diversity index were selected to reflect the changes in landscape characteristics in different buffer zones. Landscape fragmentation indexes, such as the patch density index (PD) and maximum patch index (LPI), describe the impact of human activities on landscape patches. The landscape shape index (LSI) is used to measure patch shape and connectivity. A landscape aggregation index, such as the contagion index (CONTAG), is used to describe the degree of agglomeration or spreading trend of different patch types in the landscape. Finally, landscape diversity indices, such as the Shannon diversity index (SHDI) reflect the richness and heterogeneity of the landscape. We used the landscape pattern analysis software Fragstats4.2 to calculate the landscape pattern index. The specific calculation method and ecological meaning of the landscape pattern index can be found in the relevant references.

### 2.3.3. ESV Estimation

Ecosystem Service Value estimation is an important method for measuring the quality of the regional ecological environment [39]. Based on the vector data of land-use types in the PRD in 1990, 2000, and 2017, we used the ecosystem coefficient correction method developed by Xie et al. [15] to calculate ESVs. Considering the actual situation of the PRD, the grain production in this study adopted the method of "regional correction based on farmland" to revise the equivalent factor table at the city scale [40]. According to the "Statistical Yearbook of Guangdong Province," the average grain yield of the PRD from 1990 to 2017 was 5302.61 kg/hm². In 2017, the unit price of grain purchase in the PRD was estimated to be 2.91 CNY/kg based on the grain price statistics of the South China Grain Network Trading Center. The economic value of grain produced per unit area in the PRD per year was 2224.54 CNY/hm². Built-up land is mainly economic, and its ecological value is 0. Finally, we obtained the ESV coefficient table suitable for the study area (Table 1) and calculated the ESVs of the PRD by combining Table 1 and Formula (1).

$$ESV_k = \Sigma(A_k \times VC_k) \tag{1}$$

where $ESV_k$ is the value of ecosystem service of the land use type 'k'; $A_K$ (hm²) is the area of the land-use type 'k'. $VC_k$ (CNY/hm²) is the value coefficient of the land use type 'k'.

**Table 1.** Ecosystem service value per unit of land-use type in the PRD (CNY/hm²).

| | Woodland | Grassland | Cropland | Waterbody | Unused Land |
|---|---|---|---|---|---|
| Food | 697.39 | 908.72 | 2113.31 | 760.79 | 42.27 |
| Raw material | 6297.67 | 760.79 | 824.19 | 507.20 | 84.53 |
| Gas regulation | 9129.51 | 3169.97 | 1521.59 | 5093.08 | 126.80 |
| Climate regulation | 8601.18 | 3296.77 | 2049.91 | 28,635.39 | 274.73 |
| Water supply | 8643.45 | 3212.24 | 1627.25 | 28,402.93 | 147.93 |
| Waste treatment | 3634.90 | 2789.57 | 2937.51 | 30,431.71 | 549.46 |
| Soil formation and retention | 8495.52 | 4733.82 | 3106.57 | 4205.49 | 359.26 |
| Biodiversity protection | 9531.04 | 3951.90 | 2155.58 | 7798.12 | 845.33 |
| Recreation and culture | 4395.69 | 1838.58 | 359.26 | 9911.44 | 507.20 |
| Total | 59,426.36 | 24,662.36 | 16,695.17 | 115,746.15 | 2937.51 |

## 3. Results

### *3.1. The Spatial Distribution of Landscape Pattern and ESVs in the PRD Region*

3.1.1. Spatial Distribution and Quantitative Changes in Land Use

From 1990 to 2017, the land-use space in the PRD changed significantly. Cropland and built-up land were the two land-use types showing the most dramatic changes (Figure 2). From 1990 to 2017, built-up land was primarily concentrated in the areas along the Pearl River Estuary, cropland was mainly distributed around built-up land, and woodland was mostly distributed in the eastern and western wings of the PRD. From 2000 to 2017, the spatial change in land use in the PRD showed a radial diffusion state. The land-use change in the inner and outer cities of the PRD tended to be balanced, but the intensity of land-use changes in inner cities was still higher than that in outer cities. In general, the areas with the fastest land-use changes were the central cities along the Pearl River Estuary, while the outer cities showed little change in land-use space.

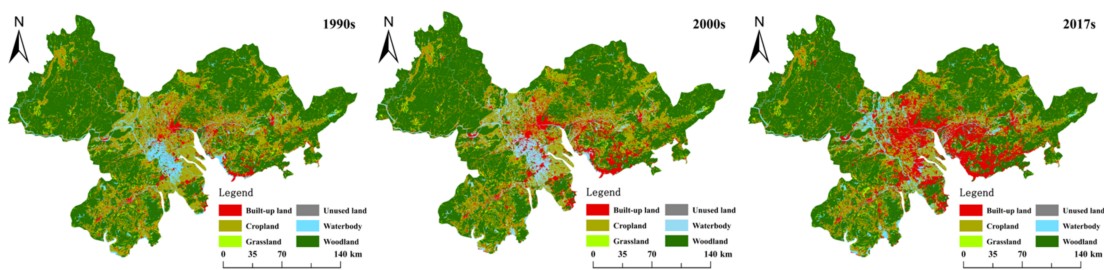

**Figure 2.** Spatial distribution of land use.

This paper uses ArcGIS10.3 software to calculate the land-use conversion relationship in the PRD region from 1990 to 2017 (Figure 3). The area of land-use change in the PRD from 1990 to 2017 amounted to 7093.36 km$^2$, accounting for 12.81% of the total land area. These results are presented in detail in Tables 2–4. The land types with the most dramatic changes were cropland and built-up land. The cropland area decreased from 15,945.92 km$^2$ in 1990 to 12,125.17 km$^2$ in 2017, and the total area decreased by 3820.74 km$^2$. Cropland was mainly created from waterbody and woodland, and the transferred areas were 464.54 km$^2$ and 110.33 km$^2$, respectively (Figure 3). Built-up land expanded rapidly. The built-up land area increased by 1204.18 km$^2$ from 1990 to 2000, and 3529.55 km$^2$ from 2000 to 2017, with a net increase of 4733.72 km$^2$, mainly from cropland, woodland, and waterbodies. The transferred areas were 2067.12 km$^2$, 923.53 km$^2$, and 598.36 km$^2$, respectively. The woodland area decreased by 17,865.82 km$^2$ from 66,096.33 km$^2$ in 1990 to 48,230.51 km$^2$ in 2017. The area of grassland, waterbody, and unused land changed little, decreasing from 1146.14 km$^2$, 4003.54 km$^2$, and 22.29 km$^2$ in 1990 to 1120.41 km$^2$, 3724.62 km$^2$, and 6.50 km$^2$ in 2017, respectively. The extent of the reduction in each case was 25.73 km$^2$, 278.91 km$^2$, and 15.79 km$^2$, respectively. The grassland area was primarily created from a woodland area of 176.23 km$^2$. The waterbody area was mainly converted to cropland and woodland, with a total converted area and ratio of 894.88 km$^2$, 26.35%, and 1.8%, respectively. The unused land was mainly converted to built-up land, with a converted area of 5.35 km$^2$, accounting for 32.97% of the total converted area.

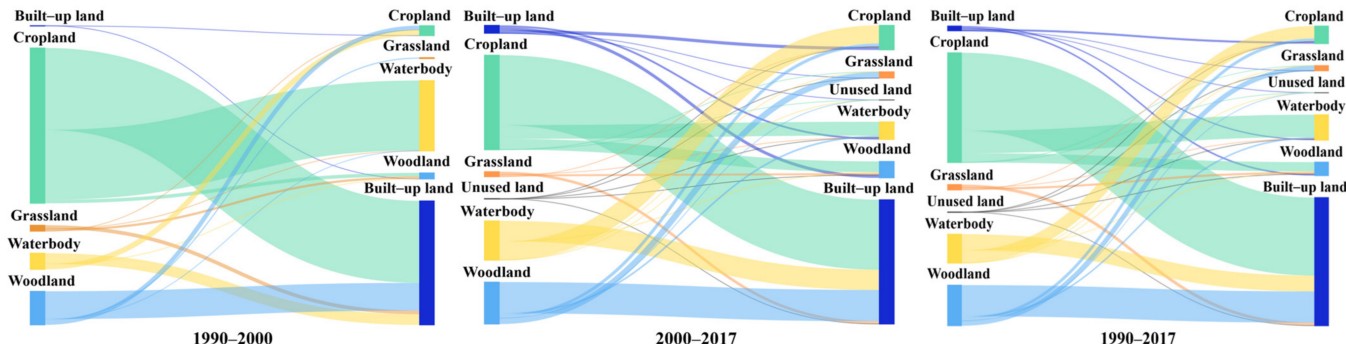

**Figure 3.** Land use change in the PRD region.

**Table 2.** Land-use transitions in the PRD between 1990 and 2000 (km²).

| Land-Use Type in 2000 | Land-Use Type in 1990 | | | | | | |
|---|---|---|---|---|---|---|---|
| | **Built-Up Land** | **Cropland** | **Grassland** | **Unused Land** | **Waterbody** | **Woodland** | **Total** |
| Built-up land | 2876.43 | 774.08 | 34.56 | 0.00 | 100.38 | 261.38 | 4046.82 |
| Cropland | 2.92 | 13,458.88 | 0.83 | 0.00 | 49.96 | 40.92 | 13,553.51 |
| Grassland | 0.00 | 0.00 | 1030.14 | 0.00 | 0.00 | 9.72 | 1039.86 |
| Unused land | 0.00 | 0.00 | 0.00 | 16.23 | 0.00 | 0.00 | 16.23 |
| Waterbody | 0.00 | 659.28 | 0.37 | 0.00 | 3162.27 | 5.04 | 3826.95 |
| Woodland | 0.48 | 32.84 | 21.24 | 0.00 | 3.21 | 29,588.15 | 29,645.92 |
| Total | 2879.82 | 14,925.07 | 1087.13 | 16.23 | 3315.82 | 29,905.20 | 52,129.28 |

**Table 3.** Land-use transitions in the PRD between 2000 and 2017 (km²).

| Land-Use Type in 2017 | Land-Use Type in 2000 | | | | | | |
|---|---|---|---|---|---|---|---|
| | **Built-Up Land** | **Cropland** | **Grassland** | **Unused Land** | **Waterbody** | **Woodland** | **Total** |
| Built-up land | 3815.02 | 2067.12 | 77.52 | 5.35 | 598.36 | 923.53 | 7486.89 |
| Cropland | 91.64 | 10,758.52 | 7.02 | 2.27 | 536.34 | 89.19 | 11,484.98 |
| Grassland | 1.66 | 4.11 | 888.22 | 0.01 | 4.34 | 176.23 | 1074.57 |
| Unused land | 0.05 | 0.10 | 0.00 | 5.62 | 0.22 | 0.33 | 6.32 |
| Waterbody | 54.83 | 406.44 | 9.73 | 1.99 | 2656.70 | 49.94 | 3179.63 |
| Woodland | 87.33 | 317.10 | 57.35 | 0.99 | 23.82 | 28,406.25 | 28,892.84 |
| Total | 4050.53 | 13,553.39 | 1039.85 | 16.23 | 3819.78 | 29,645.46 | 52,125.23 |

**Table 4.** Land-use transitions in the PRD between 1990 and 2017 (km²).

| Land-Use Type in 2017 | Land-Use Type in 1990 | | | | | | |
|---|---|---|---|---|---|---|---|
| | **Built-Up Land** | **Cropland** | **Grassland** | **Unused Land** | **Waterbody** | **Woodland** | **Total** |
| Built-up land | 2699.81 | 2911.90 | 109.88 | 5.35 | 590.50 | 1165.66 | 7483.10 |
| Cropland | 74.30 | 10,825.15 | 8.38 | 2.27 | 464.54 | 110.33 | 11,484.98 |
| Grassland | 1.48 | 4.09 | 881.32 | 0.01 | 4.48 | 183.20 | 1074.57 |
| Unused land | 0.05 | 0.12 | 0.00 | 5.62 | 0.20 | 0.33 | 6.32 |
| Waterbody | 44.55 | 837.79 | 10.25 | 1.99 | 2227.93 | 57.11 | 3179.63 |
| Woodland | 59.46 | 345.90 | 77.29 | 0.99 | 20.99 | 28,388.10 | 28,892.74 |
| Total | 2879.64 | 14,924.96 | 1087.12 | 16.23 | 3308.64 | 29,904.74 | 52,121.34 |

### 3.1.2. Changes in Landscape Index and ESVs

Figure 4 shows the changes in the four landscape pattern indices in the PRD from 1990 to 2017. At the landscape level, both LPI and CONTAG gradually decreased over time; LPI decreased from 40.83% in 1990 to 40.8% in 2017, a decrease of 0.03%, and CONTAG decreased from 68.30% in 1990 to 67.15% in 2017, a decrease of 1.15%. The overall landscape connectivity of the PRD region has declined, and the fragmentation of the landscape has deepened. Meanwhile, the patch density (PD) and SHDI showed a continuous upward trend during the study period. The PD in the PRD rose from 0.20 in 1990 to 0.22, in 2017, an increase of 0.02, and SHDI increased by 0.05 from 1.15 in 1990 to 1.20, indicating that the overall landscape diversity in the PRD had increased.

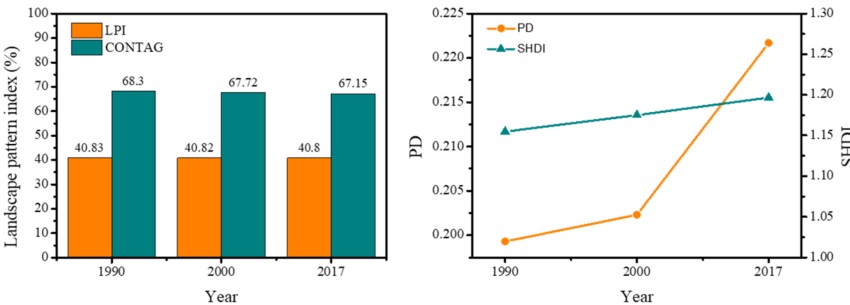

**Figure 4.** Changes in landscape index in the PRD region from 1990 to 2017.

According to formula (1), we estimated the ESVs and their changes in different land-use types in the PRD (Table 5). $4.68 \times 10^{11}$ CNY, $4.69 \times 10^{11}$ CNY, and $3.53 \times 10^{11}$ CNY in 1990, 2000, and 2017, respectively. The net value decreased by $1.16 \times 10^{10}$ CNY, a reduction ratio of approximately 24.72%. From the ESV coefficient table (Table 1), waterbodies provided the highest ESVs among the five land-use types, followed by woodland. The PRD had a vast woodland area, and woodland contributed the most to the total ESV, accounting for approximately 81.26% to 83.82%. The ESVs of woodland dropped from $3.93 \times 10^{11}$ CNY in 1990 to $2.87 \times 10^{11}$ CNY in 2017, a decrease of 27.03%. The reduction of cropland and waterbody areas also exacerbated the loss of ESVs in the PRD. The ESVs of cropland and waterbodies decreased from $2.66 \times 10^{10}$ CNY and $4.63 \times 10^{10}$ CNY in 1990 to $2.02 \times 10^{10}$ CNY and $4.31 \times 10^{10}$ CNY in 2017. It decreased by $6.38 \times 10^9$ CNY and $3.23 \times 10^9$ CNY, respectively. The ESV contribution of unused land was the smallest, from $6.55 \times 10^6$ CNY in 1990 to $2.00 \times 10^6$ CNY in 2017, a decrease of $4.64 \times 10^6$ CNY, and the proportion of total ESV value was not more than 0.01%.

**Table 5.** ESVs in the PRD region from 1990 to 2017.

| | Ecosystem Service Value (Billion Yuan) | | | Variation (Billion Yuan) | | | Rate of Change (%) | | |
|---|---|---|---|---|---|---|---|---|---|
| Land use type | 1990 | 2000 | 2017 | 1990–2000 | 2000–2017 | 1990–2017 | 1990–2000 | 2000–2017 | 1990–2017 |
| Cropland | 26.62 | 24.18 | 20.24 | −2.44 | −3.94 | −6.38 | −9.16 | −16.29 | −23.96 |
| Woodland | 392.79 | 390.31 | 286.62 | −2.47 | −103.70 | −106.17 | −0.63 | −26.57 | −27.03 |
| Grassland | 2.83 | 2.71 | 2.76 | −0.12 | 0.06 | −0.06 | −4.28 | 2.13 | −2.25 |
| Waterbody | 46.34 | 52.20 | 43.11 | 5.86 | −9.09 | −3.23 | 12.64 | −17.41 | −6.97 |
| Unused land | 0.01 | 0.01 | 0.00 | 0.00 | 0.00 | 0.00 | 0.00 | −70.85 | −70.85 |
| Total | 468.58 | 469.41 | 352.74 | 0.83 | −116.67 | −115.85 | 0.18 | −24.85 | −24.72 |

### 3.2. Spatial Distribution and Changes of Landscape Pattern and ESVs

3.2.1. Landscape Index and ESVs Spatial Change Based on Road Buffer Zone

Overall (Figure 5), the PD, LPI, and LSI indices in the buffer zones of expressways, provincial roads, and county roads in the PRD changed significantly, while the three indexes of national roads did not change much. The PD of expressways, national roads, provincial roads, and county roads in the PRD (Figure 5c,f,i,l) reached the maximum values of 0.03, 0.01, 0.03, and 0.05 at the buffer distances of 2 km, 4 km, 2 km, and 2 km, respectively. The LPI and LSI of all levels of roads maintained a relative gap and showed a downward trend. The LPI of each grade of the road reached the maximum values of 99.53%, 31.86%, 91.61%, and 26.78% at 1 km, 9 km, 1 km, and 4 km, respectively, and then the fluctuation decreased. The LSI peaked at 50.46, 31.83, 57.07, and 59.94 at the 3-, 2-, 3-, and 3-km buffer zones of various grades of roads, respectively. At the same time, the LSI of each grade of the road reached its extreme value and then decreased. From 1990 to 2017, the ESVs of all grades of roads decreased with an increase in distance. The ESVs of national roads, provincial roads, and county roads increased from 1 to 2 km and then decreased after reaching the extreme value at 2 km. In 2017, the ESVs of the three at 2 km were $8.70 \times 10^9$ CNY, $3.20 \times 10^{10}$ CNY, and $3.58 \times 10^{10}$ CNY.

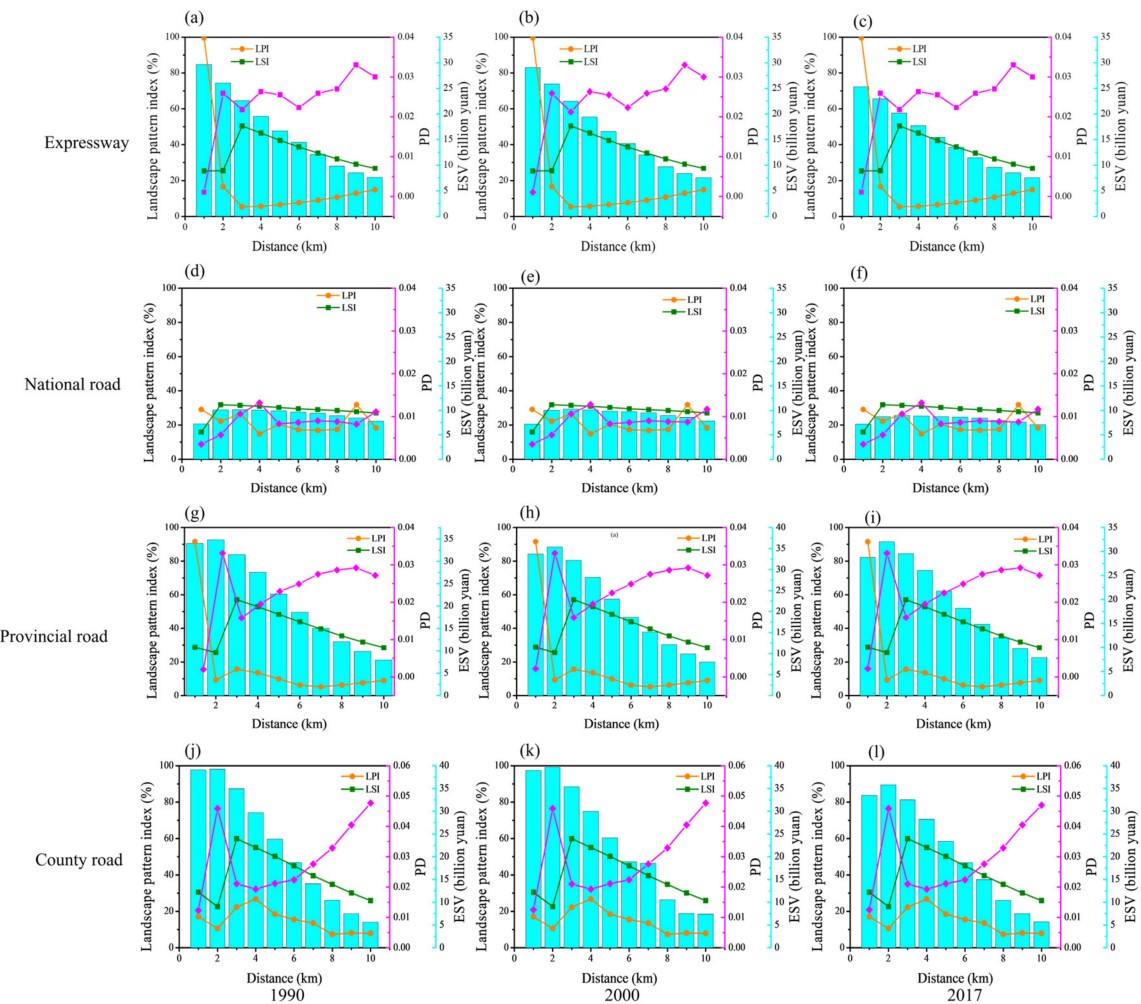

**Figure 5.** Landscape index and ESVs anges in (**a–c**) expresschway buffer zone, (**d–f**) national road buffer zone, (**g–i**) provincial road buffer zone, (**j–l**) county road buffer zone.

### 3.2.2. Spatial Change Based on Expressway Entrance and Exit and Train Station Buffer Zone

The PD of the expressway entrances and exits buffer zones in the PRD region from 1990 to 2017 rose fastest at 1 to 2 km (Figure 6a–c), reaching a maximum value of 0.03, at 1 km, and slowly decreasing after 3 km. However, the LPI and LSI did not change significantly, and both showed a slow upward trend. The reflection was most intense at 1–3 km. The two indexes increased from 1.95% and 12.20 at 1 km to 28.41% and 30.01 at 2 km, respectively. They dropped to 2.40% and 20.98 at 3 km, and the changes tended to be stable. From 1990 to 2017, the total ESV of the expressway entrances and exit buffer zones increased with an increase in distance, with the largest increase at 1–5 km. In 2017, the ESVs of the expressway entrances and exits buffer zones increased from $1.04 \times 10^{10}$ CNY at 1 km to $7.66 \times 10^{10}$ CNY at 5 km, an increase of 63.08%, totaling $6.62 \times 10^{10}$ CNY.

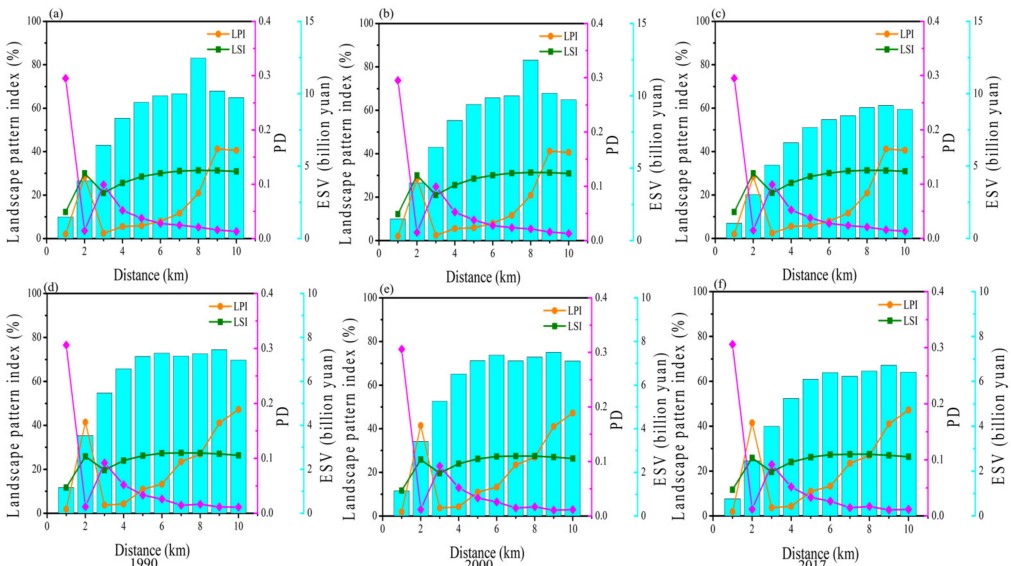

**Figure 6.** Landscape index and ESVs changes in (**a**–**c**) expressway entrance buffer zone, (**d**–**f**) exit and train station buffer zone.

The PD, LPI, and LSI indices in the train station buffer zones in the PRD fluctuated greatly from 1990 to 2017 (Figure 6d–f). In 2017, PD reached a peak value of 0.31, at 1 km, and then fluctuated and declined. The LPI and LSI rose rapidly from 1 to 2 km, rising from 1.82% and 11.67 to 41.45% and 25.93, respectively, and dropped to their lowest values at 3 km, 3.64%, and 19.61, respectively. After 3 km, the LSI changes tended to be stable. From 1990 to 2017, the total ESV in the railway station buffer zone fluctuated and increased with increasing distance. The ESVs at 1–5 km increased significantly. In 2017, the ESVs of the railway station buffer zones increased from $7.55 \times 10^8$ CNY at 1 km to $6.09 \times 10^9$ CNY at 5 km, an increase of $5.33 \times 10^9$ CNY.

### 3.2.3. Landscape Index and ESVs Spatial Change Based on River Buffer Zones

From 1990 to 2017, the PD in the river buffer zones in the PRD increased with the increased distance from 0.01 at 1 km to 0.76 at 8 km (Figure 7). The LPI reached the maximum at 1 km, reaching 95.35%, dropped to its lowest value at 2.17% at 3 km, and then showed an upward trend. The LSI rose fastest at 1–2 km, increasing from 43.72 to 81.51, and then decreased after 2 km. The ESVs in the river buffer zones decreased significantly with increasing distance and reached a peak within 1 km. The total ESVs at 1 km in 1990, 2000, and 2017 were $1.10 \times 10^{11}$ CNY, $1.12 \times 10^{11}$ CNY, and $1.04 \times 10^{11}$ CNY, respectively. The total amount of ESV within 3 km was relatively high and dropped to below $4.15 \times 10^9$ CNY after 5 km.

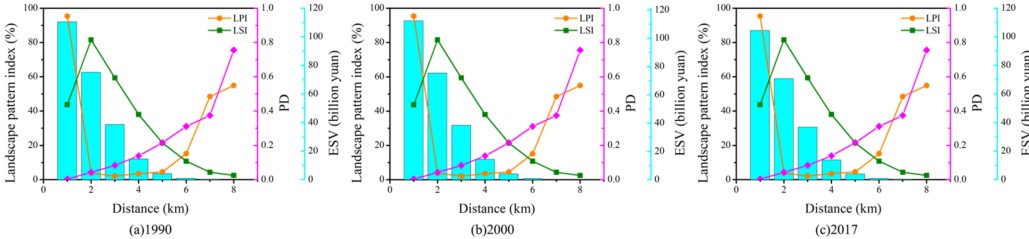

**Figure 7.** Landscape index and ESVs changes in (**a**–**c**) river buffer zone.

### 3.2.4. ESVs Spatial Change Based on City Buffer Zone

According to the "Notice on Adjusting the Criteria for Urban Size Classification" issued by the State Council of China, Guangzhou and Shenzhen are the megacities in

the PRD with permanent populations of more than 10 million, Foshan and Dongguan are megacities with populations of 5–10 million each, and Zhongshan, Huizhou, Zhuhai, Jiangmen, and Zhaoqing are megacities with populations of 1–5 million each. Figure 8 shows that from 1990 to 2017, the ESVs of the central buffer zones of nine cities in the PRD generally increased with an increase in distance, and the urban population size and ESVs were negatively correlated. From 2000 to 2017, the change rate of urban ESVs in nine cities with increasing distance slowed down, and the ESVs stabilized. Guangzhou and Shenzhen ESVs showed a fluctuating upward trend, and both grew slowly within the 3 km buffer zone of the city center, below $1.82 \times 10^7$ CNY, and then increased rapidly (Figure 8a,b). The ESVs of Foshan and Dongguan showed a gradual increase with the increase in distance, and the ESVs of the two cities dropped sharply in 2017 (Figure 8c,d). The changing trends of ESVs in Zhongshan, Huizhou, Jiangmen, and Zhaoqing were roughly the same, and the ESVs increased as the distance from the city center increased.

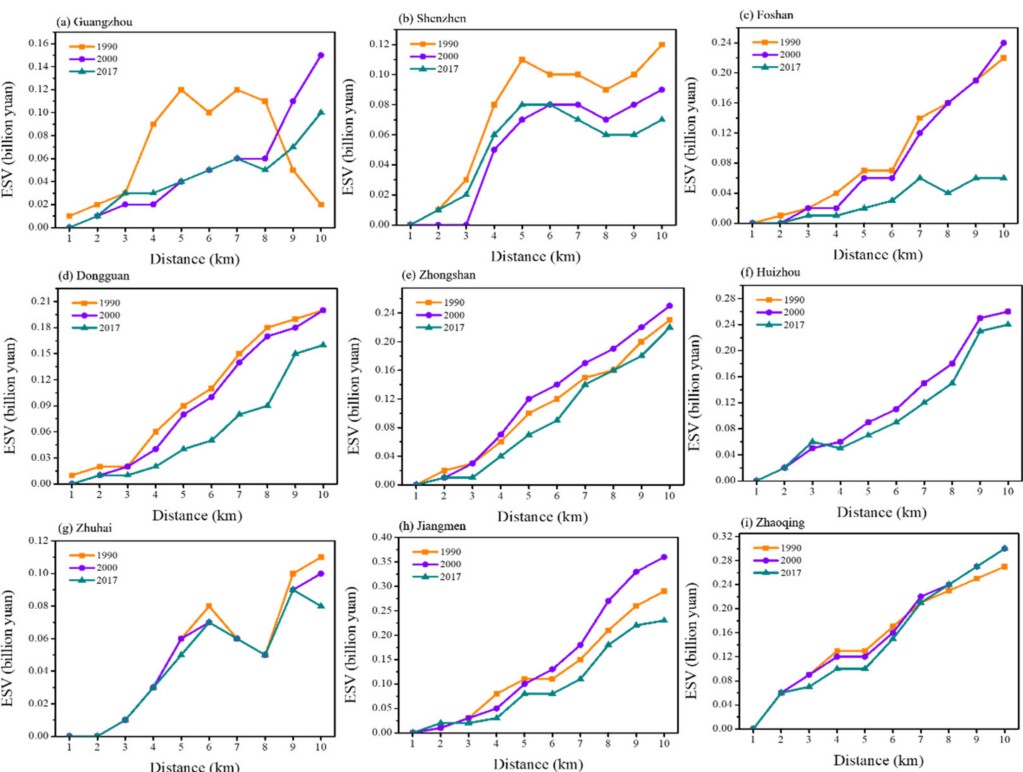

**Figure 8.** Changes of ESVs in (**a**) Guangzhou, (**b**) Shenzhen, (**c**) Foshan, (**d**) Dongguan, (**e**) Zhongshan, (**f**) Huizhou, (**g**) Zhuhai, (**h**) Jiangmen, (**i**) Zhaoqing.

## 4. Discussion

### 4.1. Urbanization and Industrialization Changed the Land Use Landscape Pattern and ESVs

China began implementing its reform and opening policy in 1978. The early urbanization of the PRD region was driven mainly by rural industrialization and opening-up policies. With its geographical advantages adjacent to Hong Kong and Macau and low land costs, the PRD region opened up an industrial development model of "Store in the front, factory in the back." To minimize the economic cost of land use and meet the demand for expanded reproduction, foreign investments in 1990–2000, especially the labor-intensive manufacturing industries in Hong Kong and Macau, were mainly distributed in villages and towns with lower land costs [41]. These industries occupied a large area of cropland, woodland, and waterbody, greatly changing the rural landscape [42]. After China joined the WTO in 2000, the PRD region responded to the country's call and actively developed foreign trade, and export-oriented urbanization developed rapidly. However, the global financial crisis that broke out in 2008 and severely affected the export-oriented economy of

the PRD. To cope with the challenges of the financial crisis, industrial enterprises in the PRD region have undergone industrial transformation and upgrading, and the industrial layout has shifted from rural areas with low land costs to urban surroundings with relatively complete facilities [43]. Now, the distribution of industrial land is more compact and intensive. Due to their geographical advantages, the cities along the Pearl River Estuary have concentrated built-up land, and the degree of land development and utilization is higher than that of the cities on the east and west wings of the PRD. Before 2000, the PRD region had extensive land use, and industrial production caused fragmentation of landscape patches and increased landscape heterogeneity [44]. After 2000, the PRD region optimized the industrial structure, and land-use changes tended to be more balanced. From 1990 to 2017, a large amount of ecological land was lost in the PRD; 2911.91 km$^2$ of cropland, 1165.67 km$^2$ of woodland, and 590.48 km$^2$ of waterbody were converted into built-up land. It is clear that urbanization and industrialization affect land-use landscape patterns and ESVs in the PRD, and the protection of urban ecosystems has been severely challenged [45].

### 4.2. The Scale of Urban Space Significantly Affects ESVs

The four cities of Guangzhou, Shenzhen, Dongguan, and Foshan have attracted inward migration and investment by virtue of their superior geographical location and have become the core areas of the PRD urban agglomeration. The core area has the closest economic ties, further accelerating the circulation of the regional population, capital, and other resource elements. The remaining five cities are in the fringe area of the PRD and have a relatively low development intensity [29]. Land-use change is the main factor leading to the reduction of the value of ecological services in metropolitan areas. The ecosystem service functions under different land-use types are quite different, and different land-use types will lead to differences in the value of ecosystem services. In this study, it is manifested as the change of land landscape type, which has an impact on the value of ecosystem services. The distribution of high-density buildings in the central urban area erodes the ecological space, and the high-density population and high-intensity human activities weaken the supply of ecosystem services [46]. The impact of the ecosystem and its service value on the landscape pattern of land use is fed back to the human socio-economic system through environmental, economic, and social impacts, which in turn affects human land-use patterns, landscape pattern configuration, utilization planning, and related policies and systems, and ultimately lead to changes in the landscape pattern of land use. The underlying surfaces of cities with high development intensity changed more drastically. The increase in impervious surfaces led to a continuous decrease in ecological land and the shrinking of woodland and waterbody areas. From 2000 to 2017, the economic development of the urban agglomeration in the PRD became mature, the speed of urbanization generally slowed, and the government increased the protection of the ecological red line. Urban land has been protected by government policies, and the ecological environment has improved to a certain extent [47].

### 4.3. The Siphon Effect of Traffic Stations Affects the Landscape and Ecology

Since 1990, the PRD has gradually changed from a dense urban area to an urban agglomeration, and the urban spatial structure has evolved into regional integration [48]. The integration of transportation infrastructure construction has been a breakthrough for regional integration in the PRD. As the largest plain area in Guangdong, the interior of the PRD is alternately distributed with hills, platforms, and alluvial plains. The area of hills and platforms accounts for one-fifth of the total area of the Pearl River Delta. Coupled with the long coastline of the PRD, the superior geographical conditions help promote the construction of the road network. With the improvement of urban functions and the continuous strengthening of regional connections, many industrial enterprises tended to spread out along expressway entrances and exits and surrounding areas, which aggravated the complexity of the surrounding land-use landscape pattern. Railway stations and express-

ways improved traffic accessibility in the surrounding areas. Under the influence of spatial agglomeration effects [49], improved traffic accessibility has increased the attractiveness of the areas around railway stations and expressway entrances and exits. The rise in land prices and population agglomeration promote the development and use of land and change the landscape pattern of land use in the PRD [50]. Within 1~3 km of railway stations and expressway entrances and exits, industrial, commercial and living residential functions are gathered, personnel activities are more frequent, and LPI and LSI respond most violently. The high-speed rail and the increase in operating mileage of expressways have increased the intensity of artificial interference in areas along the roads [51]. The landscape shape tended to be discrete. In addition, the ESVs of the railway stations, expressway entrances, and exit buffer zones increased with an increase in distance, indicating that the strong siphon effect of the traffic stations attracts a large number of people and logistics. The intensity of land development is high [52], and a large amount of ecological land in the station buffer has been converted into land for economic development, which has resulted in a decline in ESVs.

### 4.4. Differences in Highway Grades and Development Activities around Rivers Affect Regional Landscape Ecology

Roads play a leading role in the connection between urban and rural areas in the PRD. The roads pass through different ecosystems, invade the original habitat, linearly cut the landscape, destroy the regional ecosystem along the route; they also cause the density of landscape patches to increase and the patch shapes become more complex [53]. From 1990 to 2017, the rate of change of land use around roads of various grades in the PRD accelerated, and the degree of development increased [54]. The farther away from the road, the less the landscape is affected by the road, and the landscape shape tends to be regular. We observed that high-grade highways, such as expressways and national roads were mainly distributed in linear form, while county roads had a grid-like distribution connecting multiple landscape types in urban and rural areas, and the landscape pattern indices were generally higher than those of other grades of roads. This trend is mainly because the route planning of expressways often requires careful field surveys to avoid ecologically fragile areas as much as possible. After construction is completed, the surrounding areas are greened. The focus of low-grade road construction and planning is often the only convenience. A lower level of environmental protection and efficiency has caused the low-grade road network to conflict with the local land use designated by urban planning, and the cutting effect on the regional landscape is greater than that of high-grade roads [55]. Due to the acceleration of urban-rural integration and population growth in the PRD, more land is needed for industrial and commercial activities and residential area construction. Natural waterways will inevitably become semi-natural or disappear completely, and the river network structure of the PRD tends to become a backbone [56]. The ESVs along the river buffer zones of the PRD decreased with increasing distance. However, when urbanization reaches more than 30%, the density of river networks increases with the increase in urbanization level, from filling and occupying river land to protecting and restoring river networks [57]. The decline rate of ESVs in the PRD river buffer zones has slowed.

### 4.5. Uncertainty

The study has several limitations. The first is the problem of data collection. Due to the difficulty of multi-source data collection and processing, this research period does not include the most recent data. Second, ESVs are affected by multiple variables, and the spatial scale will also greatly affect ecosystem services and their valuation. The ESVs evaluation in this study is only based on the ecosystem service value equivalent table proposed by Xie [15]. The value coefficient of construction land is zero, ignoring the possible negative effects of pollution, which may affect the accuracy of the results. Finally, landscape pattern analysis still has limitations in characterizing complex urban forms and development trends (such as polycentric cities and counter-urbanization). Finally, the

delineation of the buffer zone is at a more subjective stage, and the theoretical support is insufficient. Next, we can further explore how to delimit the buffer zone radius scientifically.

## 5. Conclusions

Based on the land-use data from 1990, 2000, and 2017, this study used landscape index calculation and ESV evaluation methods to access the landscape pattern and ESV changes in the PRD, as well as the landscape pattern and ESV spatial differentiation characteristics in the multi-ring buffer zones of typical transportation infrastructure and rivers in the PRD. The study results are summarized as follows: (1) Rapid urbanization and industrialization have caused drastic changes in urban land use along the Pearl River estuary, and urban land changes on the east bank of the Pearl River are greater than those on the west bank of the Pearl River. Changes in built-up land have always been dominant and have grown steadily at the expense of agriculture and forests. The net ESVs in the Pearl River Delta region decreased by $1.16 \times 10^{10}$ CNY from 1990 to 2017. The diversity of landscapes in the PRD has increased, the connectivity of the landscape has declined, and the degree of fragmentation has increased. (2) Population growth and urban expansion continue to expand the scale of urban space. The scale of the cities in the PRD was negatively correlated with ESVs. With the development of urban land, ESVs show an increasing trend from the city center to the periphery. Within the PRD city area, the key areas for ecological protection and restoration within the city limits should be central urban areas and suburbs. (3) The regional integration process of urban development in the PRD has promoted the construction of transportation infrastructure. The siphoning effect of the railway stations and expressway entrances and exit buffer zones was strong and the landscape tends to be discrete. (4) Different grades of roads in the PRD have different effects on the regional landscape and ESVs. County roads have a greater interference effect than expressways, national roads, and provincial roads; the ESVs of the river buffer zones decrease with the increase in distance, and the river network density increases with an increase in the level of urbanization.

**Author Contributions:** Conceptualization, R.Y.; methodology, R.Y.; formal analysis, B.Q.; invetigation, B.Q.; data curation, B.Q.; visualization, B.Q.; writing—original draft preparation, B.Q.; writing—review and editing, R.Y., B.Q. and Y.L. All authors have read and agreed to the published version of the manuscript.

**Funding:** This work was supported by financial support from the Key-Area Research and Development Program of Guangdong Province (2020B0202010002), National Natural Science Foundation of China (41871177, 42171193, 41801088).

**Data Availability Statement:** The data are not publicly available for privacy reasons.

**Conflicts of Interest:** The authors declare no conflict of interest.

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
