# Peer review of "Assessment of the Impact of Land Use Change on Spatial Differentiation of Landscape and Ecosystem Service Values in the Case of Study the Pearl River Delta in China"

_land, doi:10.3390/land10111219_

Round 1
Reviewer 1 Report
Review of Effect of Land-Use Changes on Spatial Differentiation of Landscape and Ecosystem services value from the Perspective of Geographical Proximity: A case of Pearl River Delta in China
Due to estimating the value of ecosystem services, the paper covers an adequate topic. Fragmentation of land use and landscape is leading to a loss of ecosystem services, which may be exacerbated by climate change.
The paper still has few flaws, mainly structural issues, and regarding its lengthy and wordy style it should be significantly changed, and it is especially true for the Materials and Methods and the Conclusion. The base environment descriptions are missing as climate, soil, vegetation backgrounds value and historical changes at this region. The authors tried to show recent changes without environment background information.
Although the paper is statistics oriented, this is appeared at the visualization. The title of the paper implies more Geography, spatial analysis and validation of spatial analysis maps at Guangzhou, Shenzhen, Dongguan, Foshan, Zhuhai, Huizhou, Zhongshan, Jiangmen, and Zhaoqing sites.
The chapter Conclusion is insufficiently ‘conclusive’. Broader implementation of the results is really needed there and need for a list of potential applications.
General comment:
Authors should modify the title “Effect of Land-Use Changes on Spatial Differentiation of Landscape and Ecosystem services value from the Perspective of Geographical Proximity: A case of Pearl River Delta in China”. It is too long. The correct form would be: “Assessment of the impact of land use change on spatial differentiation of landscape and ecosystem service values in the case of study the Pearl River Delta in China. “.
You should modify the keywords with more specific words (e.g.): temporal variation.
Introduction
The introduction has sufficient size when it compared to the other chapters. In spite of the important citations of relevant articles from other countries, the readers thought that it could be only a local problem.
Material and method
The main problem with material methods is that the important information and simple insufficient construction of elementary descriptions are missing.
The following descriptions are missing: What is the typical soil at the sample sites? What was the classification of the soil type at sample site? (WRB and USDA soil classification). What is the description of soil profiles? What was the soil textural type? What is the climate in this
region?
Result and discussion
The current value of the natural background information won't be evident for the researchers and unfortunately this part of the chapters will not be understandable for everyone in this state. Have been spatial distribution of landscape degradation? Are there the same in both places? Can you find the orientation of degradation? Could you find a connection between degradation of landscape elements and ecosystem services values?
The figures, especially the Diagrams are not readable in this form; the font size is too small and there are different font styles: bold and normal. The authors should remake these Diagrams and maps.
The references contain too little/ just a few - international examples. It has to increase with additional 10 more international articles.
Specific comments and questions:
Line (L.) 59: need more international references
L. 49 - 59: in this part elements of the ecosystem services have to be mentioned.
L. 124 - 159: What is the international soil classification of study site? What is the typical riverine vegetation at Pearl River delta? What is the climate in this region?
L. 133. How many dollars is 7.58 × 10 12 CNY? It could be more international to convert USD currencies from CNY. Please use full sentences instead of acronyms at first time.
Conclusion
Unlike the other chapters of the paper; is way too short and does not directly flows from the results and goals. Secondly, how it could be implemented and compared for other land use areas? What sort of areas should be analyzed with that presented method? In order to use your results, list areas where your results may be potentially implemented or employed. The authors have to show the environmental background.
My final opinion is: I am going to accept after the minor revision (corrections to minor methodological errors and text editing).
Author Response
Point 1: Authors should modify the title “Effect of Land-Use Changes on Spatial Differentiation of Landscape and Ecosystem services value from the Perspective of Geographical Proximity: A case of Pearl River Delta in China”. It is too long. The correct form would be: “Assessment of the impact of land use change on spatial differentiation of landscape and ecosystem service values in the case of study the Pearl River Delta in China. “You should modify the keywords with more specific words (e.g.): temporal variation.
Response 1: We revised the title and keywords - line 1-3, 30
General comment:
Point 2: Introduction: The introduction has sufficient size when it compared to the other chapters. In spite of the important citations of relevant articles from other countries, the readers thought that it could be only a local problem.
Response 2: We have added more literature from European and American countries and reviewed their research progress
Point3: Material and method: The main problem with material methods is that the important information and simple insufficient construction of elementary descriptions are missing.
Response 3: In this section, we added the terminology information of the landscape index and explained the purpose of the buffer method
Point 4: The following descriptions are missing: What is the typical soil at the sample sites? What was the classification of the soil type at sample site? (WRB and USDA soil classification). What is the description of soil profiles? What was the soil textural type? What is the climate in this region?
Response 4: We have added a description of the climate and typical soils in the Pearl River Delta region, but this research does not involve soil research, so there is not much description of soil classification, soil profile and soil structure
Point 5: Result and discussion:The current value of the natural background information won't be evident for the researchers and unfortunately this part of the chapters will not be understandable for everyone in this state. Have been spatial distribution of landscape degradation? Are there the same in both places? Can you find the orientation of degradation? Could you find a connection between degradation of landscape elements and ecosystem services values?
Response 5: We have increased the environmental background of the Pearl River Delta region. The relationship between the degradation of landscape elements and the value of ecosystem services has been clarified in the article – line 463-470
Point 6: The figures, especially the Diagrams are not readable in this form; the font size is too small and there are different font styles: bold and normal. The authors should remake these Diagrams and maps.
Response 6: The resolution of the PDF manuscripts automatically generated by the system has been reduced. We recreated the pictures and tables. The font of the picture has been uniformly changed to New Rome.
Point 7: The references contain too little/ just a few - international examples. It has to increase with additional 10 more international articles.
Response 7: We added international literature
Specific comments and questions:
Point 8: Line 59: need more international references
Response 8: We added more international references
Point 9: L. 49 - 59: in this part elements of the ecosystem services have to be mentioned.
Response 9: We added elements of ecosystem services - line 63-68
Point 10: L. 124 - 159: What is the international soil classification of study site? What is the typical riverine vegetation at Pearl River delta? What is the climate in this region?
Response 10: We have added the above information – line 169-173
Point 11: L. 133. How many dollars is 7.58 × 10 12 CNY? It could be more international to convert USD currencies from CNY. Please use full sentences instead of acronyms at first time.
Response 11: The exchange rate of RMB against USD is different every day, it is difficult to compare, so we chose to use RMB to measure. We added the full sentences – line 176
Point 12: Conclusion:Unlike the other chapters of the paper; is way too short and does not directly flows from the results and goals. Secondly, how it could be implemented and compared for other land use areas? What sort of areas should be analyzed with that presented method? In order to use your results, list areas where your results may be potentially implemented or employed. The authors have to show the environmental background.
Response 12: We have revised the conclusion and added the environmental background of the Pearl River Delta region
Reviewer 2 Report
The theme of the paper is interesting, but it is not acceptable in the presented form.
First of all, the literature review includes most of all Chinese positions. There is a lack of more literature from European and American countries. That is why the statements in lines 98-104 are not applicable. Moreover, the literature is not up to date - about 30 from 48 positions of literature were written more than 5 years ago.
The land data comes from 1990, 2000, and 2017. At the end of the paper, the authors write that it is cased "Due to the difficulty of multi-source data collection and processing" (lines 436-438). They did not explain earlier (in section 2.2) why this situation took place and what sources caused it. If there is a lack of more present data, it could be great to present data from decades (1990, 2000, 2010, and 2020 - if there is no data it could be 2017 /lack of 3 years to 2020/). This way a study would be more interesting.
Line 165 informs about using ArcGIS software to create a buffer zone. My question is why maps with buffer zone were not included in the study?
Some of the figures are not readable (fig. 2, 3, 5, 6, 7,). There is also no information what is the description of the terms included in fig. 4 - LPI, CONTAG, PD, SHDI, and how they were calculated.
Author Response
Point 1: First of all, the literature review includes most of all Chinese positions. There is a lack of more literature from European and American countries. That is why the statements in lines 98-104 are not applicable. Moreover, the literature is not up to date - about 30 from 48 positions of literature were written more than 5 years ago.
A: We re-updated the references, added the documents of European and American countries, and revised the problem of different sentences
Response 1: We re-updated the references, added the documents of European and American countries, and revised the problem of different sentences
Point 2: The land data comes from 1990, 2000, and 2017. At the end of the paper, the authors write that it is cased "Due to the difficulty of multi-source data collection and processing" (lines 436-438). They did not explain earlier (in section 2.2) why this situation took place and what sources caused it. If there is a lack of more present data, it could be great to present data from decades (1990, 2000, 2010, and 2020 - if there is no data it could be 2017 /lack of 3 years to 2020/). This way a study would be more interesting
Response 2: Considering that the administrative divisions of the Pearl River Delta have not changed much after 2000, and the typical traffic vector data for 2020 is not available, this study could not be updated to the latest year, and only three periods of comparison were made.
Point 3: Line 165 informs about using ArcGIS software to create a buffer zone. My question is why maps with buffer zone were not included in the study?
Response 3: A: The buffer zone is only used as the basis for calculating the distance between the landscape pattern and ecosystem service value of typical features. The results obtained are presented in the form of Figures 5-8.
Point 4: Some of the figures are not readable (fig. 2, 3, 5, 6, 7,). There is also no information what is the description of the terms included in fig. 4 - LPI, CONTAG, PD, SHDI, and how they were calculated.
Response 4: The resolution of the PDF manuscripts automatically generated by the system has been reduced. The font of the picture has been uniformly changed to New Rome. In chapter 2.3.2, relevant information on landscape index terms has been added
Reviewer 3 Report
Dear author,
Thank you for the opportunity to review your article “Effect of Land-Use Changes on Spatial Differentiation of Landscape and Ecosystem services value from the Perspective of Geographical Proximity:A case of Pearl River Delta in China “ I must say, that your text is generally well written thus I have no major comments on the general structure of the article. Only a few details, mostly of formal nature, should be fixed to improve your manuscript:
- standardise the format of numerical data entry – line 135-137, 189, 199(Table 1), 218, 221,225,226,360,361
- which ArcGIS version was used to conduct analyses? - line 165
- the reference to Xie et al. is missing (probably [9] ?) at line 185
- missing superscript in units – line 189,192,226
- format tables and their captions according to the journal template – line 199, 260
- the pictures in figure 2 (line 213) are so tiny and very hard to read - I recommend to enlarge them and put them under each other on the whole page
- keep the same colors for each land-use type in figure 3 (line 233), preferably also in accordance with Figure 2
- the numbers described on lines 215-232 could be presented in the form of a new table
- check you references and present them in the requested format
With kind regards
RW
Author Response
Point 1: Standardise the format of numerical data entry – line 135-137, 189, 199(Table 1), 218,221,225,226,360,361
Response 1: We standardized the digital data entry format-lines 177-179, 249, and 258 (Table 1), 288,295,299,300, 439,440
Point 2: Which ArcGIS version was used to conduct analyses? - line 165
Response 2: ArcGIS10.3 - Line 214
Point 3: The reference to Xie et al. is missing (probably [9]?) at line 185
Response 3: We added references - Line 245
Point 4: Missing superscript in units – line 189,192,226
Response 4: We added superscript in units – line 249,241,299
Point 5: Format tables and their captions according to the journal template – line 199, 260
Response 5: We formatted the tables and its title according to the journal template – line 258, 338
Point 6: The pictures in figure 2 (line 213) are so tiny and very hard to read - I recommend to enlarge them and put them under each other on the whole page
Response 6: We recreated the picture – line 281
Point 7: Keep the same colors for each land-use type in figure 3 (line 233), preferably also in accordance with Figure 2
Response 7: We reproduced the Sankey Diagram to show the conversion relationship of land use in the Pearl River Delta – line 310
Point 8: The numbers described on lines 215-232 could be presented in the form of a new table
Response 8: We made tables 2-4 – line 306,307,308
Point 9: Check you references and present them in the requested format
Response 9: We have reorganized the references – line 575
Round 2
Reviewer 2 Report
The paper was improved.